# Frequency of Atypical Mutations in the Spike Glycoprotein in SARS-CoV-2 Circulating from July 2020 to July 2022 in Central Italy: A Refined Analysis by Next Generation Sequencing

**DOI:** 10.3390/v15081711

**Published:** 2023-08-09

**Authors:** Maria Concetta Bellocchi, Rossana Scutari, Luca Carioti, Marco Iannetta, Greta Marchegiani, Lorenzo Piermatteo, Luigi Coppola, Simona Tedde, Leonardo Duca, Vincenzo Malagnino, Lorenzo Ansaldo, Neva Braccialarghe, Stefano D′Anna, Maria Mercedes Santoro, Andrea Di Lorenzo, Romina Salpini, Elisabetta Teti, Valentina Svicher, Massimo Andreoni, Loredana Sarmati, Francesca Ceccherini-Silberstein

**Affiliations:** 1Department of Experimental Medicine, University of Rome Tor Vergata, 00133 Rome, Italy; 2Department of Oncology and Hemato-Oncology, University of Milan, 20122 Milan, Italy; 3Infectious Disease Unit, Department of Systems Medicine, University of Rome Tor Vergata, 00133 Rome, Italy; 4Department of Biology, University of Rome Tor Vergata, 00133 Rome, Italy

**Keywords:** COVID-19, variants of concern, SARS-CoV-2, spike, epidemiology, N-glycosylation

## Abstract

In this study, we provided a retrospective overview in order to better define SARS-CoV-2 variants circulating in Italy during the first two years of the pandemic, by characterizing the spike mutational profiles and their association with viral load (expressed as ct values), N-glycosylation pattern, hospitalization and vaccination. Next-generation sequencing (NGS) data were obtained from 607 individuals (among them, 298 vaccinated and/or 199 hospitalized). Different rates of hospitalization were observed over time and among variants of concern (VOCs), both in the overall population and in vaccinated individuals (Alpha: 40.7% and 31.3%, Beta: 0%, Gamma: 36.5% and 44.4%, Delta: 37.8% and 40.2% and Omicron: 11.2% and 7.1%, respectively, both *p*-values < 0.001). Approximately 32% of VOC-infected individuals showed at least one atypical major spike mutation (intra-prevalence > 90%), with a distribution differing among the strains (22.9% in Alpha, 14.3% in Beta, 41.8% in Gamma, 46.5% in Delta and 15.4% in Omicron, *p*-value < 0.001). Overall, significantly less atypical variability was observed in vaccinated individuals than unvaccinated individuals; nevertheless, vaccinated people who needed hospitalization showed an increase in atypical variability compared to vaccinated people that did not need hospitalization. Only 5/607 samples showed a different putative N-glycosylation pattern, four within the Delta VOC and one within the Omicron BA.2.52 sublineage. Interestingly, atypical minor mutations (intra-prevalence < 20%) were associated with higher Ct values and a longer duration of infection. Our study reports updated information on the temporal circulation of SARS-CoV-2 variants circulating in Central Italy and their association with hospitalization and vaccination. The results underline how SARS-CoV-2 has changed over time and how the vaccination strategy has contributed to reducing severity and hospitalization for this infection in Italy.

## 1. Introduction

In December 2019, a new virus called severe acute respiratory syndrome coronavirus 2 (SARS-CoV-2) was reported. Since its emergence, it has rapidly spread worldwide, causing the 2019 coronavirus pandemic (COVID-19) [1]. Over time, SARS-CoV-2 variants were generated showing increased transmissibility and escape from humoral immunity as well as various degrees of disease severity [2,3]. Based on their characteristics and the degree of concern raised, they have been classified as variants of concern (VOCs) and variants of interest (VOIs). In particular, VOCs have evidence in the literature of increased transmissibility, worse outcomes, and/or reduced vaccine efficacy [4,5]. VOIs have mutations in biologically significant regions (mainly in spike and in the receptor-binding domain, in nucleocapsid and ORF1ab) without clear evidence associated with increased transmissibility, virulence, and/or immune evasion [6]. Recently, the WHO added a new category to the variant tracking system, termed “Omicron sub-variants under monitoring” (VUMs); VUMs present genetic changes that are suspected to affect virus characteristics and early signals of growth advantage relative to other circulating variants but for which evidence of phenotypic or epidemiological impact remains unclear, requiring enhanced monitoring and reassessment pending new evidence [7]. Since the start of pandemic, five VOCs have been identified: B.1.1.7 Alpha variant (originally identified in the United Kingdom), B.1.351 Beta variant (originally identified in South Africa), P.1 Gamma variant (originally identified in Brazil), B.1.617.2 Delta variant (originally identified in India) and the B.1.1.529 Omicron variant (originally identified in South Africa). Currently, Omicron lineages (as pure or as recombinants) are the only variants circulating worldwide (https://gisaid.org/, accessed on 6 July 2023). To date, among Omicron descendent lineages, two recombinant VOIs (XBB.1.5 and XBB.1.16) and five VUMs (BA.2.75, CH.1.1, XBB.1.9.1, XBB.1.9.2 and XBB.2.3) require tracking (https://www.who.int/activities/tracking-SARS-CoV-2-variants, accessed on 6 July 2023).

The variants are characterized by different mutational profiles within the whole genome, but particularly in the spike glycoprotein. In detail, each VOC shows a peculiar mutational profile in spike protein (https://covdb.stanford.edu/variants/voc-comp-table/, accessed on 6 July 2023). Spike is a 1273 amino acid trimeric glycoprotein, present in the viral envelope and responsible for virus entry into host cells.

Each monomer has the region S1 (residues 1–686) with the receptor-binding domain (RBD, residues 319–541) exposed on the trimer surface, which alternates between a closed/down position and an open/up position. The region S2 (residues 687–1273) includes the fusion peptide, two heptad repeats (HR1 and HR2), a transmembrane domain and a cytoplasmic domain, promoting fusion between the viral envelope and cell membrane [8,9,10]. The virus can enter the cells when it is in the up position by binding to the human angiotensin-converting enzyme 2 (ACE2) receptor [8,11,12,13]. The part of the RBD with the residues 438–506 is referred to as the receptor-binding motif (RBM), whereas the remainder of the RBD is called the RBD core [14]. The SARS-CoV-2 RBD is the main target of neutralizing antibodies [15,16].

Viral envelope proteins are usually glycosylated to hamper recognition from neutralizing antibodies. According to the literature, twenty-two N-linked glycosylation sites were observed throughout the spike protein, differently distributed: eight glycosylation sites are found in the N terminal domain (NTD) of the S1 subunit, two are in the RBD core, three are in the carboxyl-terminus domain (CTD) and nine in S2 [17,18,19].

The role of mutations, deletions, insertions and glycosylation sites varies in the spike protein, especially when combined, and deserves rapid and further investigation in order to characterize their potential role in modulating viral transmissibility or/and recognition by endogenous or therapeutic antibodies.

Thus, by using a next generation sequencing approach, this study, which includes isolates originating from two years of the pandemic, aimed to (i) define the variants circulating in Central Italy, (ii) provide an in-depth characterization of the spike mutational profiles (focusing also on minority mutations) and (iii) investigate their correlation with clinical parameters.

## 2. Materials and Methods

### 2.1. Population Study

This retrospective study included 607 SARS-CoV-2 positive nasopharyngeal swabs (NS) obtained from individuals attending the University Hospital of Rome Tor Vergata in Central Italy from July 2020 to July 2022. At the time of enrollment, for all individuals, NS real-time PCR positive for 4 genes was available: envelope (E), nucleocapsid (N) and RNA-dependent RNA polymerase (RdRp)/spike (S) with cycle threshold (Ct) values < 35.

Real-time reverse transcription PCR Ct values were obtained by AllplexTM SARS-CoV-2 Assay Seegene (target E, N RdRp/S).

The study protocol on sample collection and sequencing of SARS-CoV-2 was approved by the Ethics Committee of Fondazione PTV Policlinico Tor Vergata (register number 46/20, 26 March 2020) and conducted in accordance with the 1964 Declaration of Helsinki. The individuals allowed viral sequencing for surveillance and/or research purpose. Demographics, epidemiological and clinical information were obtained retrospectively and collected according to the European Regulation on the protection of personal data n. 679/2016 and the Italian Legislative Decree 196/2003.

### 2.2. Viral RNA Extraction

Viral RNA was extracted from nasopharyngeal swabs with MagPure virus DNA/RNA Purification Kit (Hangzhou Bigfish Bio-tech Co., Ltd., Hangzhou, China) according to the product specifications, by using BIG FISH™ Nuetraction 32/96 Nucleic Acid Purification System. The RNA extracted was used for SARS-CoV-2 sequencing.

### 2.3. Next Generation Sequencing (NGS)

During the study period, we proceeded with NGS of the spike gene, as first, and then by whole genome SARS-CoV-2 sequencing. NGS of the spike gene was obtained by a homemade protocol, by using MiSeq platform (Illumina Inc., San Diego, CA, USA). The spike gene library search was performed according to Nextera XT DNA Library Kit (Illumina Inc., San Diego, CA, USA). Briefly, starting from amplification rounds to enriching the samples, SARS-CoV-2 RNA samples were reverse-transcribed and amplified (RT/PCR) with SuperScript III One-Step RT-PCR system for long templates (Invitrogen, Carlsbad, CA, USA) using a homemade protocol for the spike gene (forward primer [nt: 21421–21441] 5′-AGGGGTACTGCTGTTATGTCT-3′ and reverse primer [nt: 25498–25515] 5′-GGGAGTGAGGCTTGTATCGG-3′, according to NC_045512.2. When necessary, the samples were processed with an eventual second-round nested PCR (nested forward primer [nt: 21422–21444] 5′-GGGGTACTGCTGTTATGTCTTTA-3′ and nested reverse primer [nt: 25440–25462] 5′-TAGCATCCTTGATTTCACCTTGC-3′) with AmpliTaq Gold DNA Polymerase (Life Technologies Carlsbad, CA USA). For each sample, 1 ng of amplicon was involved in a tagmentation reaction by Nextera XT DNA Library Kit (Illumina Inc., San Diego, CA, USA). Finally, a unique combination of index primers (Nextera XT Index Kit v2, Illumina Inc., San Diego, CA, USA) was added to each sample according to the manufacturer’s instructions.

After purification with Ampure XP Beads (Beckman Coulter, Pasadena, CA, USA) and quantification on Qubit 2.0 Fluorometer (Life Technologies, Carlsbad, CA, USA) with Qubit dsDNA HS Assay Kit (Invitrogen, Carlsbad, CA, USA), the libraries were diluted at 4 nM as the final concentration, then they were pooled. Finally, 15 pM of the denatured pool was sequenced paired-end with MiSeq Reagent Kits v2 (2 × 250) (Illumina Inc., San Diego, CA, USA) with 6–10% of PhiX Control V3 library to monitor sequencing quality.

Whole genome sequencing of SARS-CoV-2 was obtained using the COVIDSeq Assay (96 samples) index 1 (Illumina Inc., San Diego, CA, USA) according to the manufacturer’s instructions, and 15 pM of denatured pool was sequenced paired-end with MiSeq Reagent Kit V2 (2 × 150) Illumina Inc.

### 2.4. Bioinformatics Analyses: Definition of Mutational Pattern and Related Clade, Prevalence of Mutations and Glycosylation Sites

Preliminarily, FASTQ obtained for each sample after sequencing was analyzed using the Genome Detective Virus Tool [20] to assign the taxonomic name and mutations. In addition, a quality control of the raw data obtained in the fastq format was performed by Trimmomatic [21] software in order to remove adapters, PCR primers and poor quality reads. FASTQ files were analyzed with VirVarSeq software version 1 (https://sourceforge.net/projects/virtools/files/; last accession on 6 July 2023) [22] using SARS-CoV-2 consensus (NC_045512.2) as reference. Only variants with frequency > 1% were retained for further analysis. For each sample, two consensus sequences with prevalence cutoff of 2% and 20% were generated using quasitools [23], and these were uploaded on nextstrain (https://clades.nextstrain.org, last accession on 6 July 2023) in order to assign the clade [24] and on Pangolin lineages (https://pangolin.cog-uk.io/ last accession on 6 July 2023) to assign the variant [25].

Spike mutations were defined according to the intra-prevalence as major (frequency > 90%), intermediate (frequency > 20–90%) and minor (frequency between 2–20%).

For each VOC, atypical mutations were defined as those not present in the consensus sequence of each identified variant. According to the kind of atypical mutations, all individuals were stratified in four categories: (a) individuals with major plus minor atypical (Mma) mutations, (b) individuals with only major atypical (Ma) mutations, (c) individuals with only minor atypical (ma) mutations, and (d) individuals without atypical (Wa) mutations.

The sequences obtained from all samples were subjected to in silico analysis in order to predict the presence of potential glycosylation sites by using the “N-Glycosite” algorithm (https://www.hiv.lanl.gov/content/sequence/GLYCOSITE/glycosite.html) [26]. This algorithm identifies the presence of the signal motif for glycosylation (NXS/T), which could potentially represent the target for N-glycans linking to the protein. The signal motif must begin with an asparagine (N) followed by any amino acid (aa) except proline, and the third aa residue must be a threonine (T) or serine (S). The N-GlycoSite tool marks and tallies the locations where this pattern occurs.

### 2.5. Statistical Analyses

Descriptive statistics were expressed as median values and the interquartile range (IQR) for continuous variables and the number (percentage) for categorical variables. To estimate significant differences, Fisher’s exact and chi-squared test for trend were used for categorical variables while Mann–Whitney and Kruskal–Wallis tests were used for continuous variables. Statistical analyses were performed with SPSS software package for Windows (version 23.0, SPSS Inc., Chicago, IL, USA). The Bonferroni’s correction (multiple comparison corrections) was applied in order to obtain statistically significant *p*-values.

## 3. Results

### 3.1. Patients’ Characteristics

In this study, 607 individuals infected with SARS-CoV-2 were characterized. About 95% were Italian, mainly male (N = 336, 55.4%), with a median age of 63 (IQR: 51–73) years. Regarding the severity of infection, 199 (32.8%) patients needed hospitalization, while 408 (67.2%) were non-hospitalized individuals. Of the latter, 359 attended the outpatient clinic for the administration of monoclonal antibodies against the COVID-19 disease, and 49 were SARS-CoV-2-infected individuals, without severe symptoms and/or requiring variant control for other reasons. By comparing the characteristics of hospitalized and non-hospitalized individuals, we observed that those hospitalized were older than non-hospitalized (median (IQR): 65 (54–75) vs. 62 (50–72) years, *p*-value = 0.013), more frequently males (65.3% vs. 50.5%, respectively, *p*-value = 0.002) and with a higher rate of diagnosis of pneumonia (81.4% hospitalized vs. 0.5% non-hospitalized, *p*-value < 0.001). Fewer Italian individuals were observed in the hospitalized group (176 (88.4%)) than in the non-hospitalized (399 (97.8%)), (*p*-value < 0.001).

By evaluating the interval of days from the date of first COVID-19 symptoms referred to the NS sampling date (Δ days), we found that this parameter in hospitalized patients was significantly longer than non-hospitalized individuals (median (IQR): 7 (4–10) vs. 5 (3–6) days, *p*-value < 0.001). The demographic and clinical characteristics of individuals included in the study are reported in Table 1.

Higher NS Ct values were observed in hospitalized patients (median (IQR) E-N-RdRp/S: 24 (20–27]) 22 (19–26) and 25 (21–28), respectively) compared with non-hospitalized patients (22 (19–25), 20 (18–24) and 23 (20–26), respectively, all *p*-values < 0.001).

Overall, among the 426 individuals with an available vaccination status, 298 (70%) reported to be vaccinated and had median (IQR) age of 63 (51–75) years, significantly higher than that observed in unvaccinated individuals (62 (50–71), *p*-value = 0.004).

Notably, the number of vaccinated individuals was significantly lower in the hospitalized group than non-hospitalized group (65 (39.6%) vs. 233 (88.9%) respectively, *p* < 0.001).

The group of vaccinated and hospitalized patients was significantly older than the group of vaccinated and non-hospitalized individuals, median (IQR) age 72 (61–81) vs. 62 (49–72) years, *p* < 0.001, respectively) (Table 1).

### 3.2. Characterization of Variants Circulating in the Study Population

#### 3.2.1. Temporal Characterization of SARS-CoV-2 Variants

Since July 2020, all spike sequences were characterized by D614G mutation, and until December 2020, the variants observed were B.1.177 [20E(EU1)], B.1.1.420, 20A, 19A, 20B, 20C and 20D. At the end of December 2020, the first VOIs appeared in our hospital: the first was B.1.160 variant (Portuguese variant) in one individual (0.16%), followed by the B.1.258 variant (Scottish variant) detected in three individuals (0.48%) in January–February 2021 and finally by the Eta B.1.525 variant, which appeared in April 2021 in two individuals (0.32%). The temporal distribution of variants is reported in Figure 1.

According to Italian epidemiology, the Alpha, Gamma, Beta, Delta and Omicron variants were detected from January 2021 to July 2022. In particular, Alpha B.1.1.7 variant was detected starting from January 2021 followed by Gamma P.1, detected at the end of February 2021 in a suspected transmission cluster (with an identical mutational pattern without any atypical major or minor mutations). Beta B.1.351 variant was observed in April, while Delta B.1.617.2 in June and Omicron B.1.1.529 in December 2021.

#### 3.2.2. VOCs Characterization

Overall, 553 (91.1%) individuals carried a VOC, with the Omicron variant being the most present (N = 188, 34.0%), followed by the Delta (N = 185, 33.5%) and Alpha variant (N = 118, 21.3%). The Gamma and Beta variants were detected in fewer individuals (N = 55 (9.9%) and N = 7 (1.3%), respectively). Interestingly, 31 and 26 sublineages of Delta and Omicron VOCs were identified, respectively. In Figure 2, the prevalence of Delta and Omicron sublineages are reported. In particular, three sublineages were detected with a prevalence > 10%: B.1.167.2, AY43 and AY4 for Delta and BA.1, BA.1.1 and BA.2 for Omicron VOC.

### 3.3. Characterization of Atypical Mutational Patterns

Approximately 1/3 of VOC-infected individuals (166, 30.0%) showed at least one atypical major mutation in the spike protein: twenty-seven individuals with Alpha, one with Beta, twenty-three with Gamma, eighty-six with Delta and twenty-nine with Omicron. Figure 3 shows the distribution of major and minor atypical mutations in each VOC.

Minor mutations were observed in 119 (19.6%) individuals with a median (IQR) number of 1 (1–3). Of them, 97 (81.5%) were infected by a VOC: thirty-eight by Alpha variant, twenty-nine by Omicron, fourteen by both Gamma and Delta variants and two by Beta. Overall, the significant differences in atypical major and minor variability between the VOCs are reported in Figure 3.

#### Characterization of Mma, Ma, ma and Wa Groups

To better characterize the atypical spike mutational profiles in each VOC, the population was stratified according to type of atypical mutations in four categories: individuals with major + minor atypical mutations (Mma, N = 48, of them N = 36 infected with a VOC), individuals with only major atypical mutations (Ma, N = 150, of them N = 131 with a VOC), individuals with only minor atypical mutations (ma, N = 71, of them N = 61 with a VOC) and individuals without atypical mutations (Wa, N = 338, of them N = 325 with a VOC). In Figure 4, the distribution of these atypical mutational profiles is reported for each VOC.

Interestingly, a different mutational variability was observed in each VOC. Notably, within Omicron sublineages, a lower atypical variability was observed than the other previous VOCs with higher Wa value (72.3%) compared to other Wa values observed: in Alpha (54.2%), in Beta (57.1%), in Gamma (47.3%) and in Delta (51.4%), *p*-value < 0.001 (Figure 4). Differently, a higher variability in terms of Ma was observed in Delta (41.1%) and Gamma (27.3%) versus the other VOCs (Alpha (13.6%), Beta (14.3%), Omicron (12.2%), *p*-value < 0.001 for Delta, *p*-value = 0.005 for Gamma. In terms of ma, we observed a higher and statistically significant variability only in Alpha (23.7%) versus the other three VOCs (Gamma, Delta sublineages and Omicron sublineages, *p*-value = 0.003), and not a significant increase for Beta VOC (*p*-value = 0.68), while a very low variability in Delta (2.7%) versus all the other 4 VOCs (*p*-value < 0.001) was found. Finally, the presence of Mma was low among all VOCs, with the highest prevalence observed in Gamma VOC (14.5%), *p*-value = 0.012 (Figure 4).

Overall, a significantly less atypical variability was observed in vaccinated individuals than unvaccinated individuals (Wa: 63.1% vs. 33.6%, *p*-value < 0.001, Figure 5a). In particular, a higher number of atypical minor mutations was observed in unvaccinated than vaccinated individuals (ma 21.9% and Mma 15.6% vs. ma 7.4% and Mma 5.4%, *p*-value < 0.001). Nevertheless, vaccinated people who needed hospitalization showed an increase in atypical variability compared to vaccinated people that did not need hospitalization (Wa: 46.2% vs. 67.8%, *p*-value = 0.001 Figure 5b).

Analyzing the variability in hospitalized and vaccinated individuals, a different distribution of mutations among the different VOCs was also observed, according to the vaccination status, with a statistical significance only in the Gamma VOC (*p*-value: 0.008, Figure 5c).

Looking at the virological characteristics of Mma, Ma, ma and Wa in the overall population, we observed that individuals with Mma mutations had higher Ct values than other groups (E/N/RdRp/S Ct median (IQR) Mma: 26 (22–29)/25 (22–27)/27 (24–29) vs. Ma: 22 (20–25)/21 (18–24)/23 (21–26) vs. ma: 24 (21–28)/24 (19–27)/25 (22–29) vs. Wa: 21 (19–24)/20 (18–23)/22 (20–26), all *p*-value < 0.001). Similarly, in hospitalized individuals we observed that Mma had higher Ct values than other groups: (E/N/RdRp/S Ct median (IQR) Mma: 27(24–29)/25(22–28)/28(25–30) vs. Ma: 22 (20–25)/21 (18–24)/22 (21–25) vs. ma: 25 (23–29)/25 (22–28)/25 (23–31) vs. Wa: 22 (20–25)/20 (18–24)/23 (20–26), all *p*-value < 0.001).

By evaluating the interval of days from the date of first COVID-19 symptoms to the NS sampling date (Δ days), in hospitalized patients, we found that this parameter was longer in patients with Mma than in the other groups, with a significantly longer interval when compared to the Wa group. Δ days observed in four groups of hospitalized individuals (N = 199) stratified according to the atypical mutations identified are shown in Table 2.

### 3.4. Distribution of VOCs According to Hospitalization and Vaccination Status

By evaluating the number of individuals who needed hospitalization within each VOC, significantly different rates were observed: among individuals carrying Alpha variant, 48 (40.7%) were hospitalized, within Gamma variant were 20 (36.4%), while for Delta variant were 70 (37.8%) and finally, among individuals carrying the Omicron variant, only 21 (11.2%) were hospitalized (*p*-value < 0.001) (Figure 6a).

Focusing on the 426 individuals with available an vaccination status, in each VOC, the prevalence of vaccinated individuals and hospitalized vaccinated individuals has been analyzed. The number of vaccinated individuals significantly increased with new VOCs, consequently to vaccine accessibility. Indeed, in the group of individuals carrying the Alpha variant (N = 53), only sixteen (30.2%) were vaccinated, and of these, five (31.3%) were hospitalized, and within the Gamma variant group (N = 15), nine (60.0%) were vaccinated, and of these, four (44.4%) were hospitalized; among the Delta variant (N = 141), 112 (79.4%) were vaccinated, and of these 45 (40.2%) were hospitalized, and finally in the Omicron variant (N = 175), 156 (89.1%) were vaccinated, and of these only 11 (7.1%) were hospitalized. A significantly different distribution was observed within VOCs regarding vaccination status and hospitalization in vaccinated individuals, both *p*-value < 0.001 (Figure 6b,c).

### 3.5. Characterization of Spike Mutations in the RBD Region

Besides the typical VOC spike mutations, the presence of atypical major mutations was characterized in the RBD and RBM regions (residues 319–541) in each VOC.

Within the Alpha VOC, very few individuals (N = 6, 5.1%) showed at least one atypical major mutation in the RBD and RBM regions. In detail, four mutations were observed: A348S, E484K, E484G and A522S. Within Delta, more individuals (N = 18, 9.7%) showed at least one atypical major mutation (P337S, V350A, T376I, E484D, E484K, F490L and V503I were detected in one individual, while K444I was found in five individuals, and finally A344S, G446V and N532S observed in two individuals). Within Omicron, few individuals (N = 9, 4.8%) showed atypical major mutations, more frequently as single scattered (P337L, R346T, P384L, T385I, Q474R and F490P, while S371Y was found in two individuals).

Unusually, in a single individual with a compromised hematologic clinical status carrying the BA.1 Omicron lineage, the copresence of two relevant atypical mutations (L452R and Q474R) was observed. Interestingly, later, the L452R mutation became a typical mutation in BA.4 and BA.5 Omicron lineages.

No atypical major mutations were observed in individuals with Beta and Gamma VOCs, but only the typical mutations N501Y, E484K together with K417N or K417T mutation, respectively. Table 3 shows mutational profile of RBD for each VOC.

### 3.6. Predicted Glycosylation Sites in Spike

In order to evaluate the impact of variability within the spike glycoprotein, also in terms of the presence of the 22 N-linked glycosylation sites (typical of Wuhan strain), all sequences were subjected to potential glycosylation site analysis in silico. In our population, only 5/607 sequences (<1%) showed a different N-glycosylation pattern respective to both the reference sequence of the Wuhan strain and of theirs related to VOC sublineages. In particular, within the Delta VOC, three individuals lost one potential glycosylation site (two individuals at position 717; one individual at position 1074), while one individual showed an extra atypical potential N-glycosylation site (at position 689). In the Omicron VOC, only one extra potential atypical N-linked glycosylation site was observed (one individual at position 248). Table 4 shows potential typical and atypical N-linked glycosylation sites observed in our study population.

## 4. Discussion

An in-depth characterization of the spike glycoprotein in SARS-CoV-2 circulating from July 2020 to July 2022 in Central Italy was performed. In particular, the mutational profile observed in each individual of the spike protein was characterized focusing on major and/or minor atypical mutations by comparison with the typical sequence of each VOC and by correlating with clinical parameters.

Remarkable differences have been identified due to peculiar enrichment of mutations in this region of SARS-CoV-2 during the two years. At the beginning, the individuals carried B.1.177 [20E(EU1)] with only A222V and D614G as characteristic mutations, while at the end we observed individuals carrying Omicron descendent lineages. In detail, from December 2020 to April 2021, VOIs were observed: B.1.160, B.1.258 and B.1.525 variants in one, three, and two individuals, respectively. Alpha VOC was the first one detected in our population in January 2021, followed by Gamma in February 2021 and Beta in April 2021. In our study, these three VOCs showed co-circulation, particularly in April and May 2021. Differently, starting from July 2021, the Delta VOC completely replaced the others, remaining with many lineages for six months. Similarly, Omicron, detected in December 2021, completely supplanted the previous Delta VOC (Figure 1).

In the studied population, the most frequent variant was Omicron followed by Delta (34.0% and 33.5%, respectively), both showing a great variability during their spread. In particular, 31 and 26 different sublineages were identified, respectively, even if only a few had a distribution greater than 10%: B.1.617.2, AY43, and AY4 for Delta and BA.1, BA1.1 and BA.2 for Omicron. At the end of the observation period (July 2022), Omicron VOC had five lineages, of them BA.1, BA.2, BA.4 and BA.5 were observed in our population. Currently, Omicron is the only VOC circulating worldwide (https://gisaid.org/, accessed on 6 July 2023). It continues its spread circulating with new main descendent lineages and recombinant forms XBB.1.5 and XBB.1.16. As of 3 March 2023, ECDC has de-escalated BA.2, BA.4 and BA.5 from its list of SARS-CoV-2 VOCs, as these parental lineages are no longer circulating (https://www.ecdc.europa.eu/en/covid-19/variants-concern, accessed on 6 July 2023).

In these years, the spike protein has shown an evolution, changing the mutational profile already before the emergence of the new VOCs. In fact, each VOC is characterized by its peculiar spike mutation profile, increasing the number of typical mutations from Alpha to Omicron. Compared to the original wild-type variant, the Omicron whole genome contains over 50 mutations, including 32 in the spike, that alters protein binding efficiency and immunogenicity, increasing infectivity, antibody escape ability, and the chance of reinfection [27]. Several studies analyzed the spike protein and its role in the infection/fusion with host target cells and as a target protein of vaccination-induced and monoclonal antibodies [27,28,29]. These previous studies have analyzed the crucial role of typical spike mutations. In addition, in our study, we provide an in-depth characterization of the spike mutational profiles focusing particularly on major and minor atypical mutations and investigating their correlation with clinical and virological parameters. In particular, concerning the atypical major mutations, we noted a different behavior, with an increase in general from Alpha (22.9%) to Delta (46.5%) VOC, but a dramatic decrease was observed in Omicron (15.4%). This reduction may correlate with the high number of typical mutations that characterize the Omicron spike protein compared to the other VOCs. These mutations are distributed throughout the protein, presumably stabilizing its structure and in turn reducing the virus propensity to evolve, which is in line with the fact that Omicron has been circulating since late 2021. Interestingly, a similar rate of minor and major atypical mutations was observed in individuals carrying Omicron variants. Concerning minor atypical mutations, a consistent decrease was seen from Alpha (32.2%) to Delta (7.6%) with a small increase in Omicron (15.4%), not however comparable to the values of the previous VOCs.

Focusing on RBD, the main target of neutralizing antibodies, a higher variability was found in Omicron clade with twenty-three typical amino acid variations vs. one in Alpha, two in Delta and three in Beta and Gamma. We also analyzed the atypical major mutations, and a higher intra-patient variability was observed in Delta clade with eleven atypical mutations vs. four in Alpha, zero both in Beta and Gamma, respectively and eight in the Omicron clade.

Notably, in our population, only one oncological and hospitalized individual infected by Omicron BA.1 lineage showed two atypical major mutations (L452R and Q474R) in RBD. Interestingly, the L452R is an atypical mutation in BA.1 lineage, which is involved in neutralizing activity [29], after it was observed as typical in following Omicron lineages (BA.4/BA.5).

The presence of two mutations in the same individual in this crucial region was probably consequence of the immunological condition of the patient associated with a long period of infection. In fact, the Ct relative to the E, N and RdRp/S genes were 31.6, 29.8 and 32.8, respectively and Δ day was of 6 days.

Stratifying our population according to type of atypical mutations of the spike glycoprotein (major or minor, presence or absence) and compering with virological characteristics, we found that individuals with Mma mutations had significantly higher Ct-values than other groups, all *p* < 0.001. Notably, Δ days from first COVID-19 symptoms to NS sampling was significantly longer in hospitalized patients with Mma than Wa (median (IQR) Mma: 10 (7–11) days vs. Wa: 7 (4–9) days, *p*-value = 0.005).

The presence of minor atypical mutations, associated with higher Ct-values, seems to correlate with a longer duration of infection and lower viral load, suggesting an evolution with increased variability. This may suggest that the virus, remaining longer within the host organism, acquires new minor mutations during viral replication.

Furthermore, during infection in vaccinated individuals, the virus acquired less atypical variability (more Wa, *p*-value < 0.001) than in unvaccinated subjects (where more ma and Mma were observed, *p*-value < 0.001), thus underscoring the role of vaccination/immune system action in viral evolution. However, when vaccinated individuals required hospitalization, an increase in atypical variability was observed compared to vaccinated individuals who were not hospitalized, showing virus evolution and/or acquired variability (particularly with more Ma).

Concerning infection severity, about 200 individuals (32.8%) needed hospitalization, of them 81.4% reported diagnosis of pneumonia. One hundred fifty-nine hospitalized individuals carried a VOC, and interestingly, no one had the Beta variant. This result is probably due to our dataset, including few individuals carrying Beta VOC, rather than for lack of severity of Beta.

During two years of study, we have observed that the number of hospitalizations decreased with the emergency of new VOCs, perhaps due to the increase in the number of vaccinated individuals for each VOC. In line with the literature, starting from Alpha to Omicron VOC, the rate of hospitalization significantly decreased, *p*-value = 0.005 [30,31]. However, it is important to emphasize that the infectious capacity of Omicron is mainly localized to the upper airways, thus determining a less invasive symptomatology [32].

Anti-COVID-19 vaccination in Central Italy started at the end of December 2020. Overall, the vaccinated individuals represented less than one half in hospitalized with respect to non-hospitalized and with older age *p*-value < 0.001, respectively (Table 1). To evaluate if the vaccination status and VOCs could have an impact on hospitalization, the rate of vaccinated individuals was firstly analyzed within each VOC. As the vaccination rate was significantly different between the VOCs, because it was increased over time at approximately 90% in Omicron and the rate of hospitalized individuals decreased (both *p*-values < 0.001), so the rate of vaccinated and hospitalized individuals was significantly reduced in Omicron, *p*-value < 0.001.

Finally, among the putative N-linked glycosylation sites characterized, we observed 2 potential extra typical N-glycosylation sites in all individuals with the Gamma VOC (at residues 20 and 188) and two potential extra atypical N-glycosylation sites in two individuals, one at position 689 within the Delta VOC, and one at position 248 within the Omicron BA.2 lineage. Interestingly, the residue 248 is located in the NTD, and it is known to be a target of neutralizing Abs [33]. The usage of this potential additional glycosylation site could increase the carbohydrate shield, thus masking the surface targeted by antibodies and facilitating viral evasion from B-cell mediated immune response or by therapeutic neutralizing antibodies [18,19]. Differently, the potential hyperglycosylation at residue 689, close to the furin cleavage site, might have a role in modulating viral infectivity and syncytia formation, in line with what was observed for O-glycosylation in this region [34]. However, further experiments by mass-spectrometry (MS) or by hydrophilic interaction chromatography coupled with fluorescent detection (HILIC-FLD) [35] are necessary for site-specific glycosylation profiling of SARS-CoV-2 spike proteins, in order to confirm if the identified potential N-linked glycosylation sites are actually utilized for glycosylation.

## 5. Conclusions

This study reports updated information on the temporal spread of SARS-CoV-2 variants circulating in central Italy and their association with hospitalization and vaccination.

The data provided increase knowledge about the variability of the SARS-CoV-2 spike protein in terms of both atypical mutations and glycosylation sites. Overall, in vaccinated individuals, the virus acquired less atypical variability, but when those required hospitalization, the virus evolved and acquired higher variability as major and/or minor atypical mutations. The presence of atypical minor mutations associated with higher Ct values appears to correlate with longer duration of infection and lower viral load, suggesting an evolution with greater variability.

This study underlines how SARS-CoV-2 has changed over time and how the vaccination strategy has contributed to reducing the severity and hospitalization for this infection. A lower rate of hospitalization in Omicron-infected individuals than individuals infected with other VOCs was observed, also in line with a higher vaccination rate in Italy during its emergence.

## Figures and Tables

**Figure 1 viruses-15-01711-f001:**
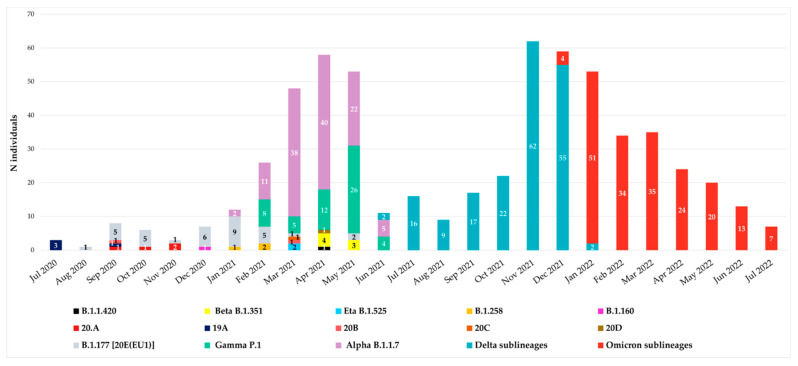
Temporal distribution of variants from July 2020 to July 2022. Monthly distribution of variants according to Nextstrain Clades and Pangolin for 607 SARS-CoV-2 Spike sequences collected from individuals attending the University Hospital of Rome Tor Vergata in Central Italy. The main waves of COVID-19 are clearly visible.

**Figure 2 viruses-15-01711-f002:**
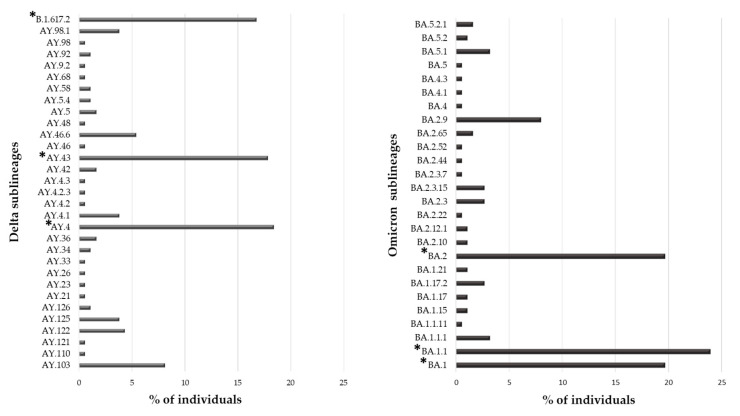
Prevalence of different sublineages in Delta and Omicron VOCs. The percentage of individuals infected with a specific sublineages was calculated in the subset of 185 and 188 individuals infected with the Delta and Omicron VOC, respectively. * Indicates sublineages with a percentage of individuals greater than 10%.

**Figure 3 viruses-15-01711-f003:**
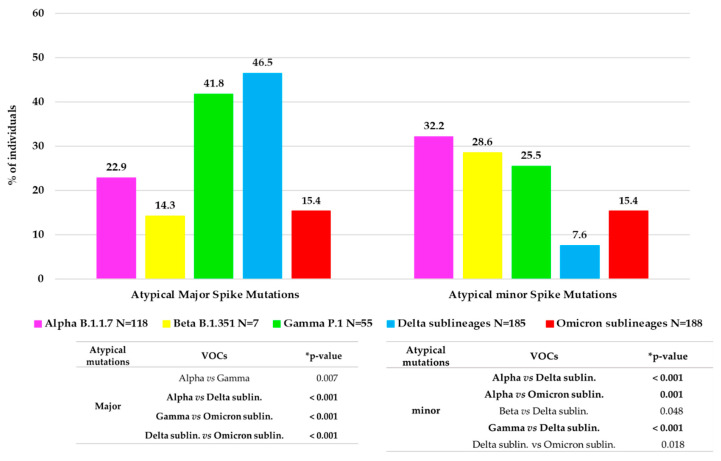
Distribution of atypical major (prevalence > 20%) and minor (prevalence > 1% and <20%) mutations observed in each VOC. The histogram reports for each VOC the percentage of individuals with at least one atypical major mutation and with at least one atypical minor mutation. * *p*-values were calculated by chi-squared test. Those remaining statistically significant after Bonferroni’s correction are reported in bold.

**Figure 4 viruses-15-01711-f004:**
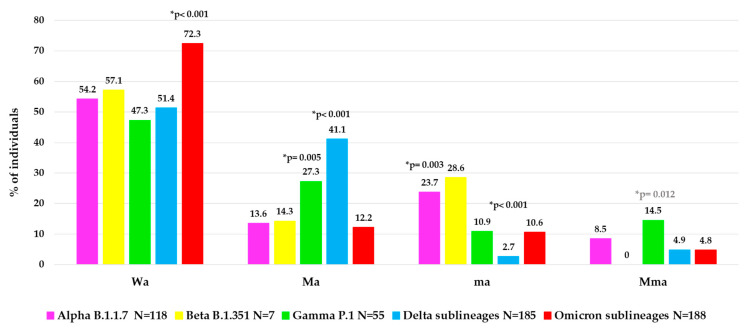
Distribution of atypical mutational profiles in spike glycoprotein observed in each VOC. Abbreviations: Wa: without *atypical* mutations, Ma: only major *atypical* mutations; ma: only minor *atypical* mutations; Mma: major + minor *atypical* mutations. * *p*-values were calculated by chi-squared test: Wa of Omicron sublineages vs. all others; Ma of Delta sublineages vs. all others, and of Gamma VOC vs. all others without Delta; ma of Delta sublineages vs. all others, and of Alpha VOC vs. all others without Beta; Mma of Gamma VOC vs. all others. *p*-values remaining statistically significant after Bonferroni’s correction are reported in black.

**Figure 5 viruses-15-01711-f005:**
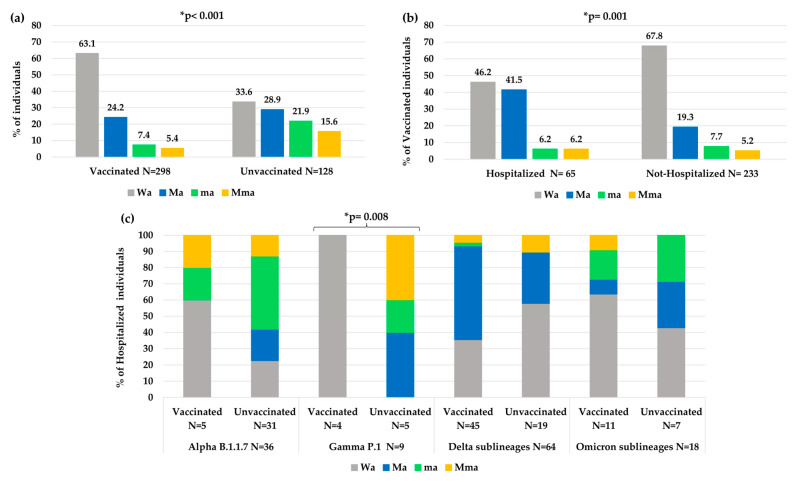
Distribution of atypical mutational variabilities according to: (**a**) vaccination status, (**b**) hospitalization status in vaccinated individuals, (**c**) vaccination status in hospitalized individuals within each VOC. Abbreviations: Mma: major + minor *atypical* mutations; Ma: only major *atypical* mutations; ma: only minor *atypical* mutations; Wa: without *atypical* mutations. * The *p*-value was calculated by chi-squared test. Those remaining statistically significant after Bonferroni’s correction are reported.

**Figure 6 viruses-15-01711-f006:**
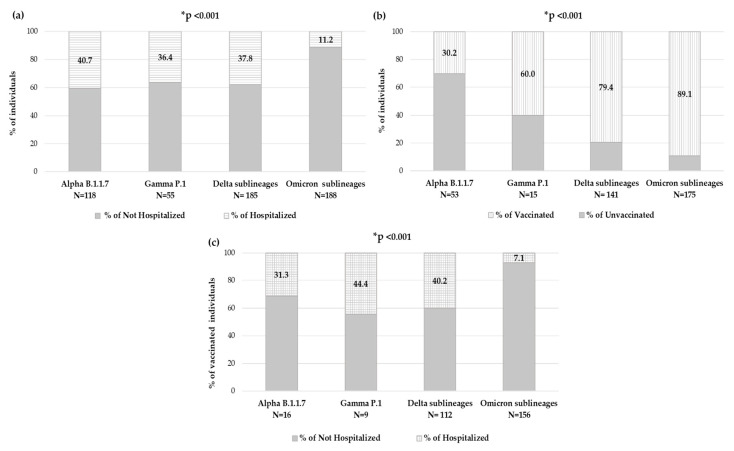
Distribution of individuals in Alpha, Gamma, Delta and Omicron VOCs according to (**a**) hospitalization status, (**b**) vaccination status and (**c**) hospitalization status in vaccinated individuals. None of the few individuals infected with the Beta VOC was hospitalized. * The *p*-value was calculated by chi-squared test, comparing Omicron sublineages vs. all others. *p*-values remaining statistically significant after Bonferroni’s correction are reported.

**Table 1 viruses-15-01711-t001:** Patients’ characteristics.

	Overall N = 607	Hospitalized N = 199	Non-Hospitalized N = 408	*p*-Value ^a^
Male, n (%)	336 (55.4)	130 (65.3)	206 (50.5)	**0.002**
*Country:*				
Italian, n (%)	575 (94.7)	176 (88.4)	399 (97.8)	**<0.001**
African, n (%)	10 (1.6)	7 (3.5)	3 (0.7)	0.017
Asiatic, n (%)	10 (1.6)	8 (4.0)	2 (0.5)	**0.003**
Hispanic, n (%)	12 (2.1)	8 (4.0)	4 (1.0)	0.024
Age, years, median (IQR)	63 (51–73)	65 (54–75)	62 (50–72)	0.013
Pneumonia, n (%)	164 (27.0)	162 (81.4)	2 (0.5) ^b^	**<0.001**
Δ day, median (IQR) ^c^	5 (4–8)	7 (4–10)	5 (3–6)	**<0.001**
*Cycle-Threshold (Ct):*				
E, median (IQR)	22 (19–25)	24 (20–27)	22 (19–25)	**<0.001**
N, median (IQR)	21 (18–24)	22 (19–26)	20 (18–24)	**<0.001**
RdRp/S, median (IQR)	23 (20–26)	25 (21–28)	23 (20–26)	**<0.001**
Vaccinated, n (%) ^d^	298 (70.0)	65 (39.6)	233 (88.9)	**<0.001**
- Male, n (%) ^e^	141 (47.3)	42 (64.6)	99 (42.5)	**0.002**
- Age, years, median (IQR) ^e^	63 (51–75)	72 (61–81)	62 (49–72)	**<0.001**

^a^ *p*-values were calculated by Mann–Whitney test or Fisher’s exact test, as appropriate. *p*-values remaining statistically significant after Bonferroni’s correction are reported in bold. ^b^ Patients hospitalized after sample collection. ^c^ The interval of days from the date of first COVID-19 symptoms referred to the NS sampling date. ^d^ Data available for 426 individuals (164 hospitalized and 262 non-hospitalized). ^e^ Data refer only to vaccinated individuals.

**Table 2 viruses-15-01711-t002:** Δ days observed in four groups of hospitalized individuals stratified according to the atypical mutations identified.

Hospitalized Patients	Δ Days	*p*-Value ^a^
Ma, N = 70	6 (4–9)	0.341 *
ma, N = 33	9 (6–10)	0.167 **
Mma, N = 28	10 (7–11)	**0.005 *****
Wa, N = 68	7 (4–9)	-

Abbreviations: Mma: major + minor *atypical* mutations; Ma: only major *atypical* mutations; ma: only minor *atypical* mutations; Wa: without *atypical* mutations. ^a^
*P*-values were calculated by Mann–Whitney test. * Wa vs. Ma; ** Wa vs. ma; *** Wa vs. Mma. *p*-values remaining statistically significant after Bonferroni’s correction are reported in bold. Data are expressed as median (IQR).

**Table 3 viruses-15-01711-t003:** Mutational profile observed in each VOC for RBD region (319–541 aa) of spike protein.

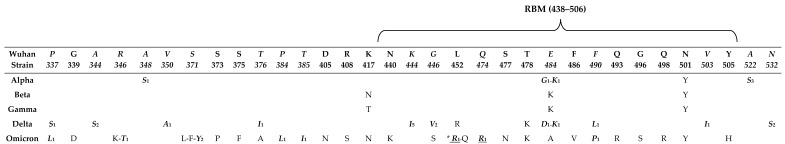

The receptor binding motif (RBM) region is below the bracket. Residues in grey are typical mutations of each VOC. Residues in black bold italic are *atypical* major mutations observed at least in a single individual; 1, 2, 3, 5 subscript indicates mutation observed in 1, 2, 3, 5 individuals. Residues underlined indicate mutations observed together in 1 individual; * indicates atypical mutation in the current lineage and then fixed in next Omicron lineages.

**Table 4 viruses-15-01711-t004:** Potential typical and atypical N-linked glycosylation sites observed in our study population.

Spike Positions	17	20 ^#^	61	74	122	149	165	188 ^#^	234	248 *	282	331	343	603	616	657	689 *	709	717 ^§^	801	1074 ^§^	1098	1134	1158	1173	1194	N-Typical Sites	N- Atypical Sites
**No-VOC** **N = 52**	52	0	52	52	52	52	52	0	52	0	52	52	52	52	52	52	0	52	52	52	52	52	52	52	52	52	22	-
**Alpha N = 118**	118	0	118	118	118	118	118	0	118	0	118	118	118	118	118	118	0	118	118	118	118	118	118	118	118	118	22	-
**Beta** **N = 7**	7	0	7	7	7	7	7	0	7	0	7	7	7	7	7	7	0	7	7	7	7	7	7	7	7	7	22	-
**Delta N = 185**	0	0	185	185	185	185	185	0	185	0	185	185	185	185	185	185	1	185	183	185	184	185	185	185	185	185	21	3
**Eta** **N = 2**	2	0	2	2	2	2	2	0	2	0	2	2	2	2	2	2	0	2	2	2	2	2	2	2	2	2	22	-
**Gamma N = 55**	55	55	55	55	55	55	55	55	55	0	55	55	55	55	55	55	0	55	55	55	55	55	55	55	55	55	24	-
**BA.1 N = 100**	100	0	100	100	100	100	100	0	100	0	100	100	100	100	100	100	0	100	100	100	100	100	100	100	100	100	22	-
**BA.2** **N = 73**	0	0	73	73	73	73	73	0	73	1	73	73	73	73	73	73	0	73	73	73	73	73	73	73	73	73	21	1
**BA.4-5** **N = 15**	0	0	15	15	15	15	15	0	15	0	15	15	15	15	15	15	0	15	15	15	15	15	15	15	15	15	21	-

^#^ Potential extra typical N glycosylation sites present only in Gamma VOC. * Potential extra atypical N glycosylation site observed. ^§^ Potential atypical N glycosylation site lost.

## Data Availability

Spike sequences were submitted to GISAID.

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
