# Peer review of "Frequency of Atypical Mutations in the Spike Glycoprotein in SARS-CoV-2 Circulating from July 2020 to July 2022 in Central Italy: A Refined Analysis by Next Generation Sequencing"

_viruses, 2023, doi:10.3390/v15081711_

Round 1

Reviewer 1 Report

This is a well written and interesting manuscript. Other than a few comments on language usage my only questions/comments are:

Have you applied multiple comparison corrections (eg Bonferroni) to your p values? Although it may not change the interpretations, it would be proper. You should include a clear statement to this effect in the statistical methods section, including a description of the method used.

Figure 5 appears to have the panels out of order. Panels A and B, which are discussed first, are placed below panel C. Typically figures identify panels in the order of discussion and if possible place them in alphabetical order from the top. Can you please put A and B at the top and C on the bottom.

Minor edits are needed to correct English grammar. Also I noted two places where the wording could be better:

Line 21 "...(NGS) was obtained from 607 individuals (298 vaccinated and/or 199 hospitalized)" is confusing because the numbers don't add up. To avoid this "...607 individuals (among them 298 vaccinated...."

Line 148: I would suggest that you use “fewer” instead of “less”. In this context the use of "less Italian" could be misread as disdain.

Author Response

Review 1

Comments and Suggestions for Authors

Have you applied multiple comparison corrections (eg Bonferroni) to your p values? Although it may not change the interpretations, it would be proper. You should include a clear statement to this effect in the statistical methods section, including a description of the method used.

Answer: Thank you for your comment, we have implemented the paragraph on statistical methods and applied the Bonferroni’s correction for multiple comparison corrections. In most cases, the Bonferroni’s correction did not change the interpretations.

Figure 5 appears to have the panels out of order. Panels A and B, which are discussed first, are placed below panel C. Typically figures identify panels in the order of discussion and if possible place them in alphabetical order from the top. Can you please put A and B at the top and C on the bottom.

Answer: I'm sorry there was an error during the file submission. The new figure is appropriate

Comments on the Quality of English Language

Minor edits are needed to correct English grammar. Also I noted two places where the wording could be better:

Line 21 "...(NGS) was obtained from 607 individuals (298 vaccinated and/or 199 hospitalized)" is confusing because the numbers don't add up. To avoid this "...607 individuals (among them 298 vaccinated...."

Answer: Thanks for the comment. Done

Line 148: I would suggest that you use “fewer” instead of “less”. In this context the use of "less Italian" could be misread as disdain.

Answer: Thanks for the comment. Done

Reviewer 2 Report

The present manuscript is a retrospective analysis of clinical data and partial characterisation of isolates of Covid patients in Italy between July 2020 and July 2022. The presence of variants of concern and occurrence of novel mutations within the Spike RBD is analysed and put into context with vaccination status and hospitalization. This is a valuable study, methodically well organized and a precedence to support pandemic preparedness notion. However, there are minor disarrangements regarding figure quality and their order (please see comments below). The language of the article would profit from a careful read-through by all authors, who will pay attention to the word order and the completeness of sentences and statements made. The findings regarding alternative glycosylation sites are at this stage an overstatement, as the authors present only the sequencing data and the derived putative N-glycosylation motifs, and this should be clearly stated. They should also add value to their findings by describing the importance of glycosylation for the behaviour of the virus more concisely (there is an extensive body of literature available), and in this view especially the physiological effect that the newly revealed altered sites could impose. Please consider the list of remarks below, which I hope you will find helpful for revision.

Title: is a good description, but unusually long, maybe “Frequency of atypical mutations in the Spike glycoprotein in SARS-CoV-2 circulating from July 2020 to July 2022 in Central Italy: a refined analysis by Next Generation Sequencing” could be used instead

Line 20: viral load is not mentioned in the article

Line 21: NGS data were obtained

Line 25: the equation symbols are better if spelled out, so more than 1 mutation

Line 25: with a distribution differing among the strains

Line 34: keywords feature their sequential numbers, which I am not sure is needed

Line 43: would profit of definition and delineation of biologically significant regions

Line 56: word order: In detail, each VOC shows a peculiar mutational profile in spike protein

Lines 58-59: word order: “the region S1 (residues 1–686) with the receptor-binding domain (RBD, residues 319-541) exposed on trimer surface”

Line 68: N-terminal domain

Line 73: “was aimed during two years”: please define: encompasses the isolates originating from two years of pandemic, or similar

Line 80: Real-Time-PCR positive results? These are 3 genes.

Line 90: MagPure is from Angen Biotech Co, Ltd.

Line 96: Library construction?

Line 106: according to

Line 109: as final concentration, then they were pooled

Lines 121-122: please cite the repositories with a website (would be enriching to the manuscript)

Line 130: potential glycosylation (not all these sites will be utilised)

Line 131: any amino acid

Line 144: were these antibodies applied because of Covid-diagnosis? Please specify.

Table 1: Patients’ characteristics

Figure 1: please increase the font of the labels, it is hardly visible

Figure 5: Figure subpanels order, c comes first, and then a and b. The legend does not fit the figure entirely.

Line 262: “significantly longer”: is this compared to all the other groups? According to the Figure, it is compared to Wa. Please state explicitly.

Line 272: significantly different rate

Figure 6: significance values could be indicated in the figure itself.

Lines 284-287: For better language, please consider wording like: In the group of individuals carrying Alpha variant…, and then Gamma variant group, etc.

Table 3. VOC abbreviation should appear one line lower, and the sequence should be labelled “S RBD wild type, Wuhan-1”, or whatever you were referring to.

Line 309: not perfectly clear: do you mean underlined means the mutation observed only in 1 individual and the fact that it is labelled with an asterisk means that it has further occurrence?

Line 315: all samples were subjected to N-glycosylation site analysis in silico – utilization of the sites or characterization of the glycosylation pattern was not really performed.

Line 318: this sentence is missing the start: within the patients infected with other variants, only 2 individuals…

Lines 322-325: The title should mention that these are putative N-linked glycosylation motifs. I do not see any underlined sequences (it might be the formatting). I do not understand how Gamma can have 24/22?

Line 340: “in our population” is better expressed as “in the studied population”

Line 346: recombinant forms

Line 360: these mutations are distributed

Line 378: comparing with virological characteristics, we found

Line 384: “undergoes on acquisition of new minor mutations”: acquires new minor mutations

Line 392: “where few individuals carrying Beta VOC”- including few individuals

Line 399: word order: “At the end of December 2020 started anti-COVID-19 vaccination in Central Italy.”- Anti-COVID-19 vaccination in Central Italy started at the end of December 2020”

Line 400. Represented less than one half

Line 406: theoretically present N-glycosylation sites

Line 422: A lower rate of hospitalization of Omicron-infected individuals than patients infected with other VOCs was observed, or similar

Author Response

The present manuscript is a retrospective analysis of clinical data and partial characterisation of isolates of Covid patients in Italy between July 2020 and July 2022. The presence of variants of concern and occurrence of novel mutations within the Spike RBD is analysed and put into context with vaccination status and hospitalization. This is a valuable study, methodically well organized and a precedence to support pandemic preparedness notion. However, there are minor disarrangements regarding figure quality and their order (please see comments below). The language of the article would profit from a careful read-through by all authors, who will pay attention to the word order and the completeness of sentences and statements made. The findings regarding alternative glycosylation sites are at this stage an overstatement, as the authors present only the sequencing data and the derived putative N-glycosylation motifs, and this should be clearly stated. They should also add value to their findings by describing the importance of glycosylation for the behaviour of the virus more concisely (there is an extensive body of literature available), and in this view especially the physiological effect that the newly revealed altered sites could impose. Please consider the list of remarks below, which I hope you will find helpful for revision.

Answer: Thank you for the overall positive comments and helpful suggestions. Accordingly, the discussion on N-glycosylation has been revised. In particular, we have discussed more extensively the mutations associated with hyperglycosylation. Furthermore, we also included a comment on the importance to confirm the predicted N-glycosylation sites with further experiments by using mass spectrometry or by Hydrophilic interaction chromatography coupled with fluorescent detection (HILIC-FLD) in order to assess their effective utilization for N-glycosylation (see lines 132-136, 424-431).

Title: is a good description, but unusually long, maybe “Frequency of atypical mutations in the Spike glycoprotein in SARS-CoV-2 circulating from July 2020 to July 2022 in Central Italy: a refined analysis by Next Generation Sequencing” could be used instead

Answer: Thanks for the suggestion. Done.

Line 20: viral load is not mentioned in the article

Answer: Thank you for this comment. In the new version of the manuscript, we have modified the sentence by specifying that viral load is expressed as ct values.

Line 21: NGS data were obtained

Answer: Done.

Line 25: the equation symbols are better if spelled out, so more than 1 mutation

Answer: Thank you. Done, for consistency with the text, we have replaced the mathematical symbol with “at least one”, a phrase used after in the manuscript.

Line 25: with a distribution differing among the strains

Answer: Done.

Line 34: keywords feature their sequential numbers, which I am not sure is needed

Answer: Thanks, we eliminated the numbers.

Line 43: would profit of definition and delineation of biologically significant regions

Answer: Done.

Line 56: word order: In detail, each VOC shows a peculiar mutational profile in spike protein

Answer: Done.

Lines 58-59: word order: “the region S1 (residues 1–686) with the receptor-binding domain (RBD, residues 319-541) exposed on trimer surface”

Answer: Done.

Line 68: N-terminal domain

Answer: Done.

Line 73: “was aimed during two years”: please define: encompasses the isolates originating from two years of pandemic, or similar

Answer: In agreement with the reviewer, in the revised manuscript we have modified the sentence.

Line 80: Real-Time-PCR positive results? These are 3 genes.

Answer: In our assay we have the information about 4 genes (E,N,RdRp,S). The sentence has been corrected.

Line 90: MagPure is from Angen Biotech Co, Ltd.

Answer: We have added the correct company name: Hangzhou Bigfish Bio.tech Co.Ltd.

Line 96: Library construction?

Answer: We have specified Spike gene library.

Line 106: according to

Answer: Done.

Line 109: as final concentration, then they were pooled

Answer: Done.

Lines 121-122: please cite the repositories with a website (would be enriching to the manuscript)

Answer: Done (see line 124).

Line 130: potential glycosylation (not all these sites will be utilised)

Answer: Done. In addition, we have remodulated the sentence according to the previous comment (see lines 132-136).

Line 131: any amino acid

Answer: Done (see line 136).

Line 144: were these antibodies applied because of Covid-diagnosis? Please specify

Answer: Done (see line 151).

Table 1: Patients’ characteristics

Answer: Done.

Figure 1: please increase the font of the labels, it is hardly visible

Answer: Done.

Figure 5: Figure subpanels order, c comes first, and then a and b. The legend does not fit the figure entirely.

Answer: Done.

Line 262: “significantly longer”: is this compared to all the other groups? According to the Figure, it is compared to Wa. Please state explicitly.

Answer: Thanks. We have rephrased the sentence (see lines 270-271).

Line 272: significantly different rate

Answer: Done (see line 281).

Figure 6: significance values could be indicated in the figure itself.

Answer: Done.

Lines 284-287: For better language, please consider wording like: In the group of individuals carrying Alpha variant…, and then Gamma variant group, etc.

Answer: We have rephrased the sentence (see lines 295-297).

Table 3. VOC abbreviation should appear one line lower, and the sequence should be labelled “S RBD wild type, Wuhan-1”, or whatever you were referring to

Answer: We have modified the table (see later).

Line 309: not perfectly clear: do you mean underlined means the mutation observed only in 1 individual and the fact that it is labelled with an asterisk means that it has further occurrence?

Answer: Yes, the asterisk on the residue L452R* indicates a mutation observed as atypical in lineage of haematological patient that has become fixed in next lineages. We have further modified the table by adding the number 1 subscript to each atypical mutation found in a single individual (this information was before missing). We also underlined the two mutations found together in a single individual with hematological problems (before was underlined only one residue).

Line 315: all samples were subjected to N-glycosylation site analysis in silico – utilization of the sites or characterization of the glycosylation pattern was not really performed.

Answer: Thank you for this comment. We fully agree with the reviewer. The sequences obtained from all samples were subjected only to in silico analysis in order to predict the presence of potential glycosylation sites by using the "N-Glycosite" algorithm (https://www.hiv.lanl.gov/content/sequence/GLYCOSITE/glycosite.html) [24]. This algorithm identifies the presence of the signal motif for glycosylation (NXS/T), that could potentially represent the target for N-glycans linking to the protein. Furthermore, as discussed in the manuscript (page 19 , lines 424-431) further experiments by mass-spectrometry (MS) or by Hydrophilic interaction chromatography coupled with fluorescent detection (HILIC-FLD) (Xie Y and Butler M. Glycobiology. 2023) are necessary for site-specific glycosylation profiling of SARS-Cov-2 spike proteins, in order to confirm if the identified potential N-linked glycosylation sites are actually utilized for glycosylation.

Line 318: this sentence is missing the start: within the patients infected with other variants, only 2 individuals

Answer: We have rephrased the sentence (see lines 330-333).

Lines 322-325: The title should mention that these are putative N-linked glycosylation motifs. I do not see any underlined sequences (it might be the formatting). I do not understand how Gamma can have 24/22?

Answer: Thanks for the comment. Accordingly, we have clarified the footnotes of the table 4, deleted the sentence of underlined sequences (it was a typo) and simplified the data inside the columns N-typical and N-atypical.

Line 340: “in our population” is better expressed as “in the studied population”

Answer: Done (see line 354).

Line 346: recombinant forms

Answer: Done (see line 360).

Line 360: these mutations are distributed

Answer: Done (see line 374).

Line 378: comparing with virological characteristics, we found

Answer: Done (see line 392).

Line 384: “undergoes on acquisition of new minor mutations”: acquires new minor mutations

Answer: Done (see line 398).

Line 392: “where few individuals carrying Beta VOC”- including few individuals

Answer: Done (see line 406).

Line 399: word order: “At the end of December 2020 started anti-COVID-19 vaccination in Central Italy.”- Anti-COVID-19 vaccination in Central Italy started at the end of December 2020”

Answer: Done (see line 413).

Line 400. Represented less than one half

Answer: Done (see line 414).

Line 406: theoretically present N-glycosylation sites

Answer: Done

Line 422: A lower rate of hospitalization of Omicron-infected individuals than patients infected with other VOCs was observed, or similar

Answer: Done (see lines 442-443).

Reviewer 3 Report

The paper "Characterization of the presence of atypical mutations in the 2 Spike glycoprotein in SARS-CoV-2 circulating from July 2020 to July 2022 in Central Italy: a refined analysis by Next Generation Sequencing" by Bellocchi and colleagues describes an overview of SARS-CoV-2 mutations across Central Italy, with the interesting addition of monitoring vaccinated and unvaccinated hosts. The study is well written and covers a sufficiently large time frame and population to warrant consideration. Moreover, the authors do not merely describe frequency of mutations, but also dig in the dataset by analyzing Spike regions affected, glycosylation sites, and stratification of mutations across different groups of patients. Overall, I would recommend acceptance after minor revision, as there are a few issues I would like to raise.

- Figure 1 is hard to read due to the low font size. This could be fixed by reducing the excessive separation between bars (which don't need to be 3D), and by sorting the bottom legend (describing the color codes assigned to variants) into more lines.

- Figure 3: the top-right legend is stretched horizontally, making it hard to read. Also, the number of significant digits in p-bvalues is different between the tests. Those with high significance (<0.00001) show that the precision of the test is to the fifth decimal value, while the less significant tests show a precision reaching only the third decimal value (0.007, 0.048, etc.). The reader is therefore led to believe that either the tests are not reporting all calculated digits (e.g. 0.00700), or that the tests are rounded, or that the actual lower limit of p-value precision is higher that that reported (so, in reality the most significant values are <0.001 and not <0.00001). Please clarify the case.

- Figure 4: the same point of the previous figure applies: please standardize and clarify the level of precision (i.e. the number of decimal digits) of the calculated p-values.

Author Response

The paper "Characterization of the presence of atypical mutations in the 2 Spike glycoprotein in SARS-CoV-2 circulating from July 2020 to July 2022 in Central Italy: a refined analysis by Next Generation Sequencing" by Bellocchi and colleagues describes an overview of SARS-CoV-2 mutations across Central Italy, with the interesting addition of monitoring vaccinated and unvaccinated hosts. The study is well written and covers a sufficiently large time frame and population to warrant consideration. Moreover, the authors do not merely describe frequency of mutations, but also dig in the dataset by analyzing Spike regions affected, glycosylation sites, and stratification of mutations across different groups of patients. Overall, I would recommend acceptance after minor revision, as there are a few issues I would like to raise.

Answer: Thanks for the comment.

- Figure 1 is hard to read due to the low font size. This could be fixed by reducing the excessive separation between bars (which don't need to be 3D), and by sorting the bottom legend (describing the color codes assigned to variants) into more lines.

Answer: accordingly, as requested we changed the figure 1.

- Figure 3: the top-right legend is stretched horizontally, making it hard to read. Also, the number of significant digits in p-bvalues is different between the tests. Those with high significance (<0.00001) show that the precision of the test is to the fifth decimal value, while the less significant tests show a precision reaching only the third decimal value (0.007, 0.048, etc.). The reader is therefore led to believe that either the tests are not reporting all calculated digits (e.g. 0.00700), or that the tests are rounded, or that the actual lower limit of p-value precision is higher that that reported (so, in reality the most significant values are <0.001 and not <0.00001). Please clarify the case.

Answer: In agreement with the reviewer, we have modified the p-values in the text, figures and tables by standardizing the p-values to the third decimal place. 

- Figure 4: the same point of the previous figure applies: please standardize and clarify the level of precision (i.e. the number of decimal digits) of the calculated p-values.

Answer: Done.